# Identification of Novel Anti-Liver Cancer Small Molecules with Better Therapeutic Index Than Sorafenib via Zebrafish Drug Screening Platform

**DOI:** 10.3390/cancers11060739

**Published:** 2019-05-28

**Authors:** Han-Syuan Lin, Yi-Luen Huang, Yi-Rui Stefanie Wang, Eugene Hsiao, Tsu-An Hsu, Hui-Yi Shiao, Weir-Torn Jiaang, Bonifasius Putera Sampurna, Kuan-Hao Lin, Ming-Shun Wu, Gi-Ming Lai, Chiou-Hwa Yuh

**Affiliations:** 1Institute of Molecular and Genomic Medicine, National Health Research Institutes, Zhunan, Miaoli 35053, Taiwan; olivia7915@hotmail.com (H.-S.L.); nmmommn@gmail.com (Y.-L.H.); stef.musicluver.girl@gmail.com (Y.-R.S.W.); eugeneychsiao@gmail.com (E.H.); boni_bt123@nhri.edu.tw (B.P.S.); khlin@nhri.edu.tw (K.-H.L.); 2Institute of Biotechnology and Pharmaceutical Research, National Health Research Institutes, Zhunan, Miaoli 35053, Taiwan; tsuanhsu@nhri.edu.tw (T.-A.H.); hyshiao@nhri.edu.tw (H.-Y.S.); wtjiaang@nhri.edu.tw (W.-T.J.); 3Division of Gastroenterology, Department of Internal Medicine, Wan Fang Hospital, Taipei Medical University, Taipei 11696, Taiwan; vw1017@yahoo.com.tw; 4TMU Research Center of Cancer Translational Medicine, Taipei Municipal Wanfang Hospital, Taipei 11696, Taiwan; gminlai@tmu.edu.tw; 5Department of Biological Science and Technology, National Chiao Tung University, Hsinchu 30010, Taiwan; 6Institute of Bioinformatics and Structural Biology, National Tsing-Hua University, Hsinchu 30013, Taiwan; 7Program in Environmental and Occupational Medicine, Kaohsiung Medical University, Kaohsiung 80708, Taiwan

**Keywords:** hepatocellular carcinoma (HCC), zebrafish drug screening, anti-angiogenesis, anti-proliferation, patient-derived xenograft

## Abstract

Hepatocellular carcinoma (HCC) ranks as the fourth leading cause of cancer-related deaths worldwide. Sorafenib was the only U.S. Food and Drug Administration (FDA) approved drug for treating advanced HCC until recently, so development of new target therapy is urgently needed. In this study, we established a zebrafish drug screening platform and compared the therapeutic effects of two multiple tyrosine kinase inhibitors, 419S1 and 420S1, with Sorafenib. All three compounds exhibited anti-angiogenesis abilities in immersed *fli1:EGFP* transgenic embryos and the half inhibition concentration (IC_50_) was determined. 419S1 exhibited lower hepatoxicity and embryonic toxicity than 420S1 and Sorafenib, and the half lethal concentration (LC_50_) was determined. The therapeutic index (LC_50_/IC_50_) for 419S1 was much higher than for Sorafenib and 420S1. The compounds were either injected retro-orbitally or by oral gavage to adult transgenic zebrafish with HCC. The compounds not only rescued the pathological feature, but also reversed the expression levels of cell-cycle-related genes and protein levels of a proliferation marker. Using a patient-derived-xenograft assay, we found that the effectiveness of 419S1 and 420S1 in preventing liver cancer proliferation is better than that of Sorafenib. With integrated efforts and the advantage of the zebrafish platform, we can find more effective and safe drugs for HCC treatment and screen for personalized medicine.

## 1. Introduction

Hepatocellular carcinoma (HCC), which accounts for 90% of primary liver cancer cases, is the fourth leading cause of cancer-related deaths worldwide [1]. East Asia and sub-Saharan Africa have higher incidence rates; however, the incidence rate is increasing in the USA [2,3]. Risk factors for HCC include hepatitis B and C virus infection, alcoholic and non-alcoholic liver diseases, iron storage diseases and aflatoxin. The risk factors affecting the HCC incidence rate have regional differences [4]. The pathogenesis of HCC is a multistep progression involving chronic inflammation, steatosis, fibrosis, cirrhosis, hyperplasia, dysplasia, and the development of a malignant tumor [5].

Preventions of HCC can be classified as primary, secondary, and tertiary. Primary prevention includes the reduction of exposure to risk factors for liver cancer and vaccine injection against hepatitis B, especially in developing countries. Secondary prevention includes curative treatments and palliative treatments. Tertiary preventions inhibit the recurrence of liver cancer [6]. Several treatment methods are available for the different stages of HCC. The curative treatment for HCC is resection in the very early stage (single), liver transplantation in the early stage (three nodules <3 cm, but no disease), and radiofrequency ablation (RFA) in the early stage (three nodules <3 cm, with disease), but most patients are in the advanced stage and cannot undergo surgery. Other palliative treatments include transarterial chemoembolization (TACE) in the intermediate stage (multi-nodular), chemotherapy, and radiotherapy. The systemic drug Sorafenib is the first line treatment in the advanced stage (portal invasion) [5,7]. Sorafenib combined with radiotherapy can increase survival time and may be used for the prevention of recurrence after resection or ablation [8]. Regorafenib and lenvatinib showed positive results in Phase III studies [9].

The safety and efficacy of Bevacizumab plus TACE were evaluated in carefully selected patients and demonstrated antitumor activity in patients with un-resectable HCC [10,11]. Drugs selectively block the specific target pathways of HCC progression, such as angiogenesis, signal transduction, and the epigenetic dysregulation of tumors [5]. Vitamin K1 plus Sorafenib synergistically inhibited HCC cell growth [12]. HCC is a highly vascularized solid tumor, so angiogenesis can be an important target for therapy [13]. The role of angiogenesis in HCC has provided systematic therapeutic means for HCC [14]. Anti-angiogenesis therapies mainly involve the blocking of vascular endothelial growth factor (Cediranib), and are sometimes combined with other angiogenic receptors, such as platelet-derived growth factor receptor (Sorafenib, Sunitinib, Linifanib, and Pazopanib) or the fibroblast growth factor receptor (Brivanib). Anti-vascular endothelial growth factor (VEGF) antibody (Bevacizumab) can block VEGF-mediated angiogenic signals [8].

Zebrafish (*Danio rerio*) is a tropical freshwater fish, and has become a popular vertebrate model in biological research, as it plays a bridging role between in vitro cell-based models and in vivo mammalian models. Zebrafish are small in size, have a large number of progeny, have ex utero development, the optically transparent embryo allows direct morphological observation, and the genomics are highly conserved to humans [15]. Zebrafish have emerged as a human disease model for toxicology, angiogenesis, DNA damage, cancer, aging, and drug screening [16,17,18,19].

The zebrafish cancer model can accelerate the discovery of therapeutic means. During zebrafish embryonic development, intersegmental vessel (ISV) formation is an angiogenic process [20]. ISVs sprout from the dorsal aorta (DA), connecting to the dorsal longitudinal anastomotic vessel (DLAV) between 24 and 48 h post fertilization (hpf) [21,22]. In this study, the embryos were immersed with drugs at 24 hpf and we observed the inhibition of ISV angiogenesis at about 50 hpf, in order to identify drugs that inhibit ISV formation for compounds which may target cancer angiogenesis.

Toxicity assays have been developed for zebrafish teratogenicity [23], cardiotoxicity, and neuro-sensory organ toxicity [24]. The liver is the organ that metabolizes drugs. Drug hepatoxicity was observed to significantly reduce both liver red fluorescent protein (RFP) intensity and liver size in LiPan *Tg(fabp10a:DsRed;elaA:egfp)* transgenic zebrafish [25]. In this study, we immersed three days post-fertilization (dpf) *Tg(fabp10a:EGFP-mCherry)* embryos with drugs for two days, and followed the above study to detect two measurable points as hepatotoxicity indicators: RFP intensity and liver size.

Zebrafish are an excellent animal model for studying liver cancer. Neoplasia can be induced by carcinogens [26,27,28]. Stable overexpression of *kras^V12^* was generated in transgenic zebrafish-induced liver tumorigenesis [29]. Pathways and genes responsible for liver development (hepatogenesis) and liver cancer formation (hepatocarcinogenesis) are largely conserved between human and zebrafish [30,31]. Zebrafish liver tumors are highly analogous to human tumors in terms of comparative analysis of microarray data and ultrasound biomicroscopy [27,28]. Therefore, using the transgenic zebrafish liver cancer model is a useful tool for HCC research and identifying new therapeutic drugs [32].

We showed that hepatitis B virus X antigen (HBx) plays an important role in hepatocarcinogenesis, causing genomic instability, activating signal pathways, and affecting the epigenomic status [31]. Using the HBx-induced HCC mouse model, we identified five common regulator genes: *Edn1*, *Bmp7*, *Bmp4*, *Spib*, and *Src* that were up-regulated in the pre-cancer stage [33]. Using transgenic zebrafish, we found that HBx induced steatosis, inflammation, and hyperplasia upon aflatoxin treatment [34]. Overexpression of *HBx* in *p53* mutant (*p53*^−^, tumor suppressor gene) zebrafish can cause HCC formation at 11-months-old. Overexpression of *src* also induced HCC at 11 months, but together with the *p53* mutation can produce earlier HCC formation at seven-months-old [34]. Our HBx-induced HCC zebrafish model is more similar to human HCC, as the animal progresses from steatosis to fibrosis, hyperplasia and dysplasia, prior to developing HCC. Our zebrafish model also shares similar molecular mechanisms with human hepatocarcinogenesis in terms of the activation of *src* and its downstream signaling pathways. This phenomenon resembles human HCC formation and provides a potential platform for in vivo drug screening for therapies for human liver cancer platforms [34]. In this work, we used *Tg(fabp10a:HBx,p53^−^)* and *Tg(fabp10a:src,p53^−^)* transgenic fish at 11- and 9-months-old injected with novel small molecules and observed the therapeutic effects in comparison to Sorafenib. We also used *Tg(fabp10a:HBx,src*,*p53*^−/+^) transgenic fish overfeeding to accelerate HCC formation at five months of age, then orally fed drugs for one month, and then observed the therapeutic effect.

Due to heterogeneity of cancers, one-treatment-fits-all medicine approaches have been replaced by a precision approach and personalized medicine [35,36]. In the genomic era, next-generation sequencing has gradually improved personalized medicine by identifying the cancer-causing variants of target genes; however, predicting the outcome simply replying on genomic data is difficult [37]. Patient-derived xenotransplantation (PDX) provides rapid screening for selecting drugs that can inhibit tumor cell proliferation and migration. Injecting human cancer cells into zebrafish embryos at 2 dpf when the immune response of zebrafish is not yet established can avoid rejection [17], and does not require prior immunosuppressive treatment [38,39]. Integrating the genomic approach, the PDX model, and a high-throughput drug screening platform will help the clinical identification of effective personalized medicine [36]. In this study, we used tumor cells labeled with CM-DiI from patients to microinject 2 dpf embryos, let the tumor grow for one day, and then treated with drugs for two days, and then observed the anti-proliferation effects of the drugs.

## 2. Results

### 2.1. Determination of Anti-Angiogenesis Effect Using Zebrafish Larva and Half Maximal Inhibitory Concentration (IC_50_)

HCC is a solid tumor with one of the most vascular structures. One of the targets of Sorafenib for advanced HCC, vascular endothelial growth factor (VEGF), is the major driver of angiogenesis. Therefore, anti-angiogenesis could provide a therapeutic means to treat HCC. Conserved genes and pathways are responsible for embryonic angiogenesis and tumor-induced angiogenesis. Embryonic angiogenesis can be easily observed in zebrafish using transgenic fluorescent zebrafish *Tg(fli1:EGFP)* embryos. Therefore, we first examined the anti-angiogenesis effects of therapeutic drugs for HCC.

Previous studies established a drug automatic high-throughput screening (HTS) method using zebrafish embryos [40,41]; we established an anti-angiogenesis platform using Vatalanib 2HCl (PTK787, VEGFR2/KDR inhibitor) as a positive control. We observed the total number of ISVs present in the trunk of the embryos and the number of complete inter-segmental vessels (ISVs), as shown in Figure 1A.

To test whether novel multiple kinase inhibitors 419S1 and 420S1 exhibit anti-angiogenesis effects, one dpf *Tg(fli1:EGFP)* embryos were incubated with various concentration of the drugs in E3 buffer (embryo water) with 0.003% phenylthiourea (PTU) for 24 h to inhibit pigmentation. The images were captured at two dpf, and analyzed for anti-angiogenesis. We observed the formation of ISVs in the trunk part of the embryo. According to the length of the ISVs, a score was assigned as 1, 3/4, 1/2, 1/4, or 0 if the ISVs grew to the full length, or 3/4, 1/2, 1/4, or no ISV growth at all, respectively (Figure 1B). As shown in Figure 1C–E, the drugs exhibited different degrees of anti-angiogenesis. Using serial dilution, the observed IC_50_ values for 419S1, 420S1, and Sorafenib were 0.7506, 3.329, and 0.5339 μM, respectively (Figure 1F). The IC_50_ for Sorafenib is close to that of 419S1.

### 2.2. 419S1 and 420S1 Exhibit Anti-Proliferation and Anti-Migration Ability in Xenotransplantation Assay

Our previous study showed that EDN1 overexpression in 293T cells (293T/EDN1) exhibited enhanced migration and proliferation [42]. To examine whether 419S1 and 420S1 can block tumor cell migration and proliferation, we injected 293T/EDN1 cells labelled with CM-Dil (red fluorescence) into 2 dpf embryonic yolk sacs of *Tg(fli1:EGFP)*. The embryos were incubated with different compounds. Cell migration and proliferation activity were observed for five days.

One day post-injection (dpi), embryos were distributed into 96-well plates and treated with 1% dimethyl sulfoxide (DMSO), 1.25 μM 419S1, 5 μM 420S1, and 2.5 μM Sorafenib, which were pre-determined by a sub-lethal dosage test. The representative images of the embryos with increased proliferation and migration behavior at 3 dpi and 5 dpi are shown in Figure 2A. Treatments of 419S1 and 420S1 significantly decreased cell proliferation compared to the DMSO control, suggesting 419S1 and 420S1 may be as potent as Sorafenib in inhibiting tumor cell proliferation (Figure 2B). Treatments of 419S1 and 420S1 significantly reduced 293T/EDN1 cell migration ability and the effects were even much better than those observed with Sorafenib treatment (Figure 2C). Together, our results indicate that 419S1 and 420S1 decrease hepatoma cell migration and proliferation in zebrafish.

### 2.3. 419S1 Exhibits Less Hepatoxicity

Gong et al. developed an in vivo hepatoxicity assay using zebrafish embryos [25]. To check whether 419S1 and 420S1 possess toxicity to zebrafish liver, we used three dpf *Tg(fabp10a:EGFP-mCherry)* embryos that show green and red fluorescence in the liver as an indicator, and treated with 1% DMSO, 0.625 μM 419S1, 2.5 μM 420S1, and 1.25 μM Sorafenib, separately. According to Gong et al., two different images were captured: one with a fixed exposure time to capture RFP intensity below saturation for intensity measurement and comparison (Figure 3A,C), and the other with sufficient exposure time to show the whole liver region for size measurement at five dpf (Figure 3B,D). The two kind of images were analyzed using Image J software (an open source Java image processing program). We found that compared to the DMSO-treated control, 420S1 significantly reduced liver RFP intensity and shrank the liver size. Sorafenib-treated embryos also showed the same hepatotoxicity, suggesting that 2.5 μM of 420S1 might be toxic to the liver (Figure 3C). However, we found that 0.625 μM 419S1 treatment produced no liver toxicity, as indicated by the RFP intensity and liver size being similar to the DMSO control.

### 2.4. 419S1 and 420S1 Exhibited Less Embryonic Toxicity Than Sorafenib

We examined embryonic toxicity with various concentrations (0.017, 0.17, 1.7, 17, and 170 µM). Morphology and survival rate were observed at different time points (1, 2, 3, 4, and 5 dpf). Using serial dilution, the observed half lethal concentration (LC_50_) values for 419S1, 420S1, and Sorafenib at 5 dpf were 10.75, 20.84, and 5.294 μM, respectively (Figure 4A). The LC_50_ for Sorafenib was the lowest, indicating it has higher embryonic toxicity. The therapeutic index (LC_50_/IC_50_) for 419S1, 420S1, and Sorafenib was 14.322, 6.260, and 9.916, respectively. 419S1 has the highest therapeutic index, indicating it is a better drug.

Small molecular compounds have less embryonic toxicity compared to Sorafenib, which causes embryo death and abnormal phenotypes including pericardial edema (red arrow), retarded yolk sac reabsorption (green arrow), retarded swim bladder inflation (black arrow), spinal curvature, and/or short body length (blue arrow). 419S1- and 420S1-treated embryos also developed these abnormalities at higher dosages (Figure 4B).

### 2.5. 419S1 and 420S1 May Prevent HCC Formation in HBx(p53^−^) and src(p53^−^) Transgenic Zebrafish

Next, we examined whether 419S1 and 420S1 prevent HCC formation using the transgenic fish HCC model. 419S1 and 420S1 are both known as novel Src inhibitors, and from our previous study, HBx induced HCC formation via activating Src expression [33,43]. Our previous study showed that overexpression of *HBx* in a *p53* mutant background caused the zebrafish to develop HCC at 11 months of age, and overexpression Src in a p53 mutant background facilitates the HCC formation at nine months of age [43]. To examine the therapeutic effect, adult zebrafish with HCC (*Tg(fabp10a:HBx,p53^−^)* at 11.3 months and *Tg(fabp10a:Src,p53^−^)* at 8.7 months) were retro-orbitally injected with 419S1, 420S1, and Sorafenib at concentrations of 15, 30, and 30 μg/g, respectively. The fish were given the drug twice a week for one month, then sacrificed to obtain the livers for histopathological and quantitative polymerase chain reaction (Q-PCR) analyses.

As our previous study showed, the expression of cell-cycle-related genes, such as *ccne1*, *cdk1*, and *cdk2* (Table 1) were elevated during HCC formation. We examined the expression of those three genes using Q-PCR. In the DMSO control group, the cell-cycle-related genes were upregulated compared to the control fish. However, in both the 419S1 and 420S1 groups, the three genes were much lower compared to the DMSO-treated group. The positive control, Sorafenib-treated fish, also exhibited lower expression of *ccne1*, *cdk1*, and *cdk2* (Figure 5A–C). These phenomena were also observed in *Tg(fabp10a:Src,p53^−^)* fish. The data suggest that 419S1 and 420S1 might be able to prevent HCC formation in the transgenic zebrafish HCC model.

We also examined the pathological features using hematoxylin and eosin staining (H&E staining). As shown previously, the progression of HCC is multistep: hepatitis, steatosis, fibrosis, cirrhosis, hyperplasia, dysplasia, and final development into HCC. According to the histopathological features of *Tg(fabp10a:HBx,p53^−^)* and *Tg(fabp10a:src,p53^−^)* [43], we judged H&E stains with an enlargement of polymorphic nuclei, prominent nucleoli, and an increased number of mitotic figures, as being diagnosed with HCC. Fish treated with 419S1, 420S1, and Sorafenib all exhibited minor histopathological features, diagnosed as normal, hepatitis, or steatosis (Figure 5D,E). Representative images are shown in Figure 5F. We also used two different lines of *Tg(fabp10a:HBx,p53^−^)* for anti-HCC validation. As shown in Figure 6, the three drugs tested had a similar effect on decreasing the expression of the cell cycle/proliferation markers.

### 2.6. 419S1 and 420S1 Prevent HCC Formation in HBx,src(p53^−^) Obese Transgenic Zebrafish HCC Model

In *Tg(fabp10a:HBx,src,p53^−^)* triple transgenic zebrafish, diet-induced obesity (DIO) accelerated HCC formation at five months of age, and increased the cancer incidence three-fold (unpublished results of our laboratory). To examine the therapeutic effect, five-month-old adult *Tg(fabp10a:HBx,src,p53^−^)*-DIO zebrafish with HCC were orally fed 419S1 and 420S1 at concentrations of 15 μg/g and 30 μg/g, respectively. Fish were fed twice a week for one month and sacrificed to obtain the livers for the histopathological and Q-PCR analyses.

Compare to the DMSO control group, both 419S1 and 420S1 treatment decreased the expressions of *ccne1*, *cdk1,* and *cdk2* (Figure 7A–C). We used immunohistochemistry analysis to measure protein expression levels of the proliferation marker, proliferating cell nuclear antigen (PCNA). Both 419S1 and 420S1 treatment dramatically decreased PCNA staining (Figure 7D). The pathological features determined using H&E stain demonstrated that five-month-old adult *Tg(fabp10a:HBx,src,p53^−^)*-DIO zebrafish developed HCC in 30% of fish and dysplasia in 40% of fish. Fish treated with 419S1 and 420S1 demonstrated dramatically reduced pathology and displayed 67% and 75% normal hepatocyte features, respectively (Figure 7E). The representative images of H&E staining and PCNA IHC are shown in Figure 7F. The data suggest that 419S1 and 420S1 prevent HCC formation in the *Tg(fabp10a:HBx,src,p53^−^)* obese transgenic zebrafish HCC model.

### 2.7. 419S1 and 420S1 Perform Better Than Sorafenib in Patient-Derived-Xenograft Model

Due to advances in bioscience, personalized treatment is the goal of clinical research for future use in clinical therapy. Personalized treatment involves profiling the genomic DNA, gene, and protein expression in either the patient’s germline DNA or tumor tissues. However, the research on drug testing does not allow direct manipulation in the patient; therefore, xenograft mice for personalized treatment is an alternative preclinical trial. However, to perform PDX in mice, one million tumor cells are required (1 cm^3^), which is rarely obtained from HCC surgery. Therefore, we searched for other substitutive animal models for the PDX experiment.

Zebrafish have emerged as an attractive animal system for modeling human cancers and are a cost-effective tool for anticancer drug screening. Tumor xenografts in zebrafish have been recently developed; the transparency of zebrafish embryos and transparent adult fish allow us to monitor tumor progression and migration behavior under a real-time confocal microscope [44,45,46]. The zebrafish cancer model can complement other cancer models, such as mouse and human models [47]. We successfully applied this technology to monitor the migration behavior of EDN1- [42], MAN1A1- [48], and RPIA-overexpressed cells [49]. We tested 419S1 and 420S1 on the PDX model in zebrafish and tested drugs’ inhibitory effects on cell proliferation. Based on drug selection using the xenograft nude mouse model, the results will affect personal treatment regimens for the clinical physician and patient.

Using the patient-derived xenograft zebrafish model, we are able to identify safe and personalized medicine for future use in clinical therapy. We used carboxyfluorescein diacetate succinimidyl ester (CFSE) to label tumor cells, and xenotransplanted to 2 dpf AB fish, gradually increasing the temperature from 28 °C to 37 °C two days after xenotransplantation. We observed the proliferation at one dpi and three dpi by taking images, and analyzing all images in ImageJ. We successfully finished tumors from patients of liver cancer patients from Wan-Fang Hospital (Figure 8). We found 419S1 reduced liver cancer proliferation for 39% of cancer patients, and 420S1 reduced tumor proliferation for 25% of cancer patients, whereas Sorafenib only reduced liver cancer proliferation for 15% of cancer patients and tumor proliferation was unchanged for 69% of cancer patients (Figure 8). This result prompted us to use PDX to determine the drug effectiveness for improving cancer survival and reduce the disease burden using individualized treatment.

## 3. Discussion

In response to the lack of effective anti-HCC drugs, we established a drug screening platform in zebrafish to quickly find novel anti-HCC drugs for targeted therapy. Our first strategy involved finding anti-angiogenic small molecules for anti-HCC treatment. Malignant tumors are always accompanied by angiogenesis to supply oxygen and nutrients for tumor growth. In particular, HCC is one of the most vascularized solid tumors; thus, anti-angiogenesis provides a potential therapeutic target. By observing inter-segmental vessels (ISVs) of the trunk part of the embryo’s body in *Tg(fli1:EGFP)* embryos with green fluorescent protein expressed in the vessels, we tested compounds for anti-angiogenic activity. Although embryonic angiogenesis and tumor-induced angiogenesis use similar pathways and molecules, zebrafish angiogenesis is not exactly the same as tumor-induced angiogenesis. In a future study, we will apply the patient-derived xenograft model and tumor-induced angiogenesis will be monitored under time-lapse microscopy, and we will test the drugs for anti-tumor induced angiogenesis directly.

Further candidate drugs were tested in adult transgenic zebrafish with HCC formation to see if they can prevent HCC formation. We established many transgenic zebrafish that developed HCC at 11 or 9 months; we established *HBx*, *Src*, and *p53* triple mutant transgenic fish with over-feeding-induced obesity, hoping to create an earlier onset and increase HCC incidence. Using both transgenic fish, we proved the effectiveness of 419S1 and 420S1 in preventing HCC formation.

We also established a xenotransplantation assay to test if the candidate drugs have anti-proliferative abilities in a high throughput manner. Using high-content imaging platforms, we enhanced the ability to screen for the anti-proliferative and anti-migratory effects of small molecules in zebrafish embryos. The culture temperature for zebrafish is 28 °C, and for mammalian cells is 37 °C. We adjusted the temperature by gradually increasing the temperature from 28 °C to 37 °C in two days, so the embryos carrying the cancer cells can survive in 37 °C for several days.

Primary tumors directly transplanted from patients into an immuno-deficient mouse model (also known as patient-derived-xenograft, PDX) has become the emerging personalized medicine model [50,51]. Previous studies revealed that tumor cell behavior is similar in zebrafish embryos compared to within the human body after xenotransplanting into zebrafish embryos, i.e., if these cells are metastatic tumor cells, they will migrate to other parts of the body of the zebrafish one to three days post-injection. Therefore, we performed a xenotransplantation experiment using patient-derived tumor cells transplanted into zebrafish embryos, and tested whether 419S1 and 420S1 inhibited tumor cell growth and metastasis, with cancer chemotherapy patients as a reference. There are some limitations in using zebrafish as a PDX model: there is little knowledge about their niche structures and micro-environmental cues, differences in size (small zebrafish vessels versus large human cells), and no adult immune-permissive zebrafish lines are available yet.

We were surprised to find both 419S1 and 420S1 are better than Sorafenib. These compounds would be useful in clinical trials for future development. We found 419S1 has the most potential as an anti-HCC drug, because 419S1 had a lower IC_50_ compared to 420S1, exhibits anti-proliferative and anti-migratory activities in the xenotransplantation assay, and most importantly, 419S1 may have less hepatotoxicity. We also observed 420S1 treatment had the highest cardiac edema rate compared to 419S1 and Sorafenib treatments. 419S1 has less cardiac edema compared to Sorafenib at 6 dpf.

We used *Tg(fabp10a:EGFP-mCherry)* embryos expressing red florescent protein in liver as a hepatotoxicity model, and observed liver RFP intensity and liver size to examine the hepatotoxicity of the new drugs. Hepatotoxicity is one of the main causes of drug attrition in the pharmaceutical industry. Zebrafish phenotypic assays showed considerable advantages in this assay. We would like to find new drugs that not only have strong anti-cancer effects but also have little effect on the liver and other normal cells of the organism. We can achieve this goal by using the zebrafish animal model.

Drug metabolism occurs primarily in the liver. The expression of *fabp10a*, which encodes for a liver fatty acid binding protein, is possibly an appropriate marker of chemical-induced hepatotoxicity. If lipid metabolism changes, hepatotoxicity is reflected in RFP intensity. When compounds produce toxicity via a non-lipid mechanism, RFP in the liver also promotes the observation and measurement of the liver size. Changes in liver size due to toxicity has severe effects on the liver, inducing hypertrophy or atrophy [25]. Molecular- or cellular-level analysis should also be conducted to determine the real cause for the hepatotoxicity of 420S1.

## 4. Materials and Methods

### 4.1. Ethics Statement

The HCC tissues were obtained from Taipei Municipal Wanfang Hospital (Taipei, Taiwan); procedures were undertaken in accordance with the Taipei Medical University Joint Institutional Review Board (TMU-JIRB, 201404018) and National Health Research Institutes Institutional Review Board (NHRI-IRB, EC1030206). All adult participants provided written informed consent and there were no child participants.

All zebrafish experiments were approved by the Institutional Animal Care and Use Committee (IACUC) of the NHRI and were in accordance with the International Association for the Study of Pain guidelines (protocol number: NHRI-IACUC-103021-A). Taiwan Zebrafish Core Facility (TZCF) at NHRI or TZeNH is a government-funded core facility, and since 2015, the TZeNH has been accredited by Association for Assessment and Accreditation of Laboratory Animal Care International (AAALAC).

### 4.2. Transgenic Zebrafish Lines

Four transgenic zebrafish lines *Tg(fli1:EGFP)*; *Tg(fabp10a:EGFP-mCherry)*, *tp53^zdf1/zdf1^*; *Tg(fabp10a:HBV-HBx-mCherry,myl7:EGFP)* i.e., *Tg(fabp10a:HBx,,p53^−^)*, *tp53^zdf1/zdf1^*; *Tg(fabp10a:src,myl7:EGFP)* i.e., *Tg(fabp10a:Src,,p53^−^)*, and the cross of those two fish generated *Tg(fabp10a:HBx,src,p53^−^)* were used in this study. The *Tg(fli1:EGFP)* containing *fli1* (friend leukemia integration 1 transcription factor, 15 kb) promoter, driving the expression of enhanced green fluorescent protein (EGFP) in all blood vessels throughout embryogenesis [52], enables anti-angiogenesis readout for drug treatment. *Tg(fabp10a:EGFP-mCherry)* contains liver-specific *fabp10a*
*(fatty acid binding protein 10a)* promoter driving the expression of EGFP and mCherry fusion proteins [53], facilitating observation of hepatotoxicity. *Tg(fabp10a:HBx,p53^−^)*, *Tg(fabp10a:Src,p53^−^)* were established by Dr. Lu in our laboratory—the fish developing HCC at 11 and 9 months, respectively [43].

### 4.3. Zebrafish Maintenance

Zebrafish (*Danio rerio*) were maintained in the Zebrafish Core Facility at NTHU-NHRI (ZeTH). The zebrafish were incubated at 28 °C under continuous flow of air in the zebrafish core facility and with automatic control of a 14 h light/10 h dark cycle. All zebrafish experiments were conducted under the approval of the Institutional Animal Care and Use Committee (IACUC) at NHRI (NHRI-IACUC-103021-A).

### 4.4. Embryos Collection

One day prior to fertilization, male and female adult zebrafish were placed individually into mating tanks with inner mesh. Male and female fish were separated by a separator and left in mating cages overnight. The next morning after the removal of the separator, the couple zebrafish stimulated by the light started to chase each other and lay eggs and sperm. After 1 h, the embryos were collected and transferred to a 100 mm dish with E3 solution (5 mM NaCl, 0.17 mM KCl, 0.33 mM CaCl_2_, and 0.33 mM MgSO_4_, pH 7.0) [54] and incubated at 28 °C for 6 h. The unfertilized and dead embryos were removed, and the remaining live embryos were replenished with fresh E3 solution and kept for incubation.

### 4.5. Angiogenesis Inhibition Drug Screening Platform

At about 24 h post-fertilization (hpf), the chorion was removed with the protease (1.5 mg/mL for 5 min) from *Streptomyces griseus* (Sigma-Aldrich Inc., St. Louis, MO, USA). The de-chorion embryos were distributed into 24-well culture plates with eight embryos per well containing E3/PTU (1-phenyl-2-thiourea, 0.003%) buffer. The *Tg(fli1:EGFP)* embryos were treated with drugs at various concentrations and resuspended in 1% DMSO. The embryos were anesthetized with tricaine (ethyl 3-aminobenzoate methanesulfonate, MS-222 (Sigma-Aldrich In., St. Louis, MO, USA) final concentration 0.016%) for about 50 hpf to prevent movement and their images were captured. The images were analyzed by measurement of the length of inter-segmental vessels to determine the anti-angiogenesis effect of individual compounds.

### 4.6. Sources of Compounds

BPR1J419S1 and BPR1J420S1 are multiple kinase inhibitors involving Src (one of Src family kinase, a family of non-receptor tyrosine kinases) and Flt-3 (Fms-related tyrosine kinase-3 is a receptor tyrosine kinase), and both can inhibit VEGFR. Sorafenib, a Src inhibitor, was used as a positive control. Sorafenib was the first FDA-approved multi-kinase inhibitor for the treatment of advanced HCC in 2007 [5]. It inhibits tumor cell proliferation and tumor angiogenesis by targeting different signaling pathways, cell proliferation (via the serine/threonine RAF kinases), and angiogenesis (via VEGFR and platelet-derived growth factor receptor (PDGFR)), significantly increasing the survival of patients with advanced HCC [55]. The compounds were provided by Dr. Tsu-An Hsu and Dr. Weir-Torn Jiaang of the Institute of Biotechnology and Pharmaceutical Research, National Health Research Institutes (Zhunan Town, Miaoli County, Taiwan). Vatalanib 2HCl (PTK787), used as a positive control for anti-angiogenesis, was purchased from Selleckchem Inc. (Houston, TX, USA).

### 4.7. Retro Orbital Injection (RO Injection)

The *Tg(fabp10a:HBx,p53^−^)* adult fish at 11.3 months old and *Tg(fabp10a:src,p53^−^)* adult fish at 8.7 months old were used to inject the drugs and observe the histopathological changes of the hepatocytes. DMSO was used as the negative control; the experimental drugs (419S1 and 420S1) and the positive control (Sorafenib) were injected with final concentrations of 15 μg, 30 μg, and 30 μg per gram of body weight, respectively. Phenol Red was added to the injection solution as a color indicator. The adult fish were injected with the same volume (1.9 μL) per fish, which weighed 0.1 g. The adult fish were anesthetized with 0.016% tricaine to prevent their movement and placed with dorsal side up and faced right on damp sponge. The injection needle was positioned with the bevel facing up such that if the fish’s eye were a clock, the needle was pointed at the seven o’clock position and at a 45-degree angle to the fish. The injection duration was twice a week for one month for a total of eight times.

### 4.8. Procedure of Diet-Induced Obesity (DIO)

DMSO, 419S1, and 420S1 treatment were applied to diet-induced obesity (DIO) *Tg(fabp10a:HBx,src,p53^−^)* fish. Three-month-old *Tg(fabp10a:HBx,src,p53^−^)* fish were divided into three groups containing 10 or 12 fish in 2 L tanks, and fed three times per day with four times over the regular amount of hatched *Artemia salina* (about 83 mg cyst/fish/day) to induce obesity. This procedure was sustained for eight weeks.

### 4.9. Oral Gavage

*Tg(fabp10a:HBx,src,p53^−^)* DIO zebrafish were oral gavaged with drugs to test the anti-tumorigenesis effect. After the diet adjustment for two months, fish were orally fed with 5 μL drug solution twice per week for a month, and then were sacrificed and livers were collected for further molecular and pathological analyses. For oral feeding, we anaesthetized the fish and placed them on a wet sponge, then fed with 5 μL drug solution using microliter syringes (Hamilton, MA, USA) and FTP-22-25 plastic feeding tubes (Instech Laboratories, Plymouth Meeting, Inc. PA, USA). The fish were put back into fresh water immediately after oral gavage to recover from anaesthetization. The dosages of 419S1, 420S1, and Sorafenib were 15, 30, and 30 μg per gram of body weight, respectively, and the average weight of fish was 425 ng. All the fish were weighed and length measured before and after the whole process of oral gavage.

### 4.10. Liver Tissue Collection and Paraffin Section

After one month of RO injection or oral gavage, the fish were sacrificed, and the livers were removed and divided into two parts for RNA isolation and paraffin section. The liver tissues were frozen in liquid nitrogen immediately after sectioning and stored at −80 °C for later RNA isolation. For histochemistry analysis, liver tissues were fixed in a 10% formalin solution (Sigma-Aldrich Inc., St. Louis, MO, USA). The fixed tissue was embedded in paraffin, and sectioned into 5-μm thicknesses mounted on poly-L-lysine coated slides, and the sections were stained with hematoxylin and eosin (H&E) stain, which was performed at the Pathology core facility.

### 4.11. Total RNA Isolation

Total RNA was isolated by NucleoSpin^®^ RNA kit (MACHEREY-NAGEL INC., Bethlehem, PA, USA). About 30 mg of tissue were collected and placed in 350 μL RA1 buffer and 3.5 μL β-mercaptoethanol (MilliporeSigma, St. Louis, MO, USA) mixture, and stored at −80 °C at this step. Upon RNA isolation, the samples were thawed slowly at room temperature and then disrupted by pestles to lyse tissue. The lysate was filtrated with NucleoSpin^®^ Filter (violet ring) by centrifuging at 11,000× *g* for 1 min to reduce viscosity and clear the lysate. After centrifugation, 350 μL of 70% ethanol prepared by DEPC water (diethyl pyrocabonate water) was added to the filtrate and mixed well by pipetting up and down. The lysate was loaded into a NucleoSpin^®^ RNA column (light blue ring) and centrifuged at 11,000× *g* for 30 s. Following, 350 μL of membrane desalting buffer was added to the column and centrifuged at 11,000× *g* for 1 min.

To each column, we added 95 μL DNAse reaction mixture containing 10% RNase-free DNase and 90% reaction buffer for DNase, and placed at room temperature for 30 min to digest the genomic DNA. After DNase digestion, 200 μL RAW2 buffer was added to the column to inactivate the DNase and centrifuged at 11,000× *g* for 30 s. Then, the columns containing RNA were transferred to new 2-mL collection tubes, and 600 μL and 250 μL RA3 buffer were added sequentially followed by centrifuged at 11,000× *g* for 30 s and 2 min to clean up the RNA samples twice. Finally, the columns were transferred into new 1.5-mL tubes that were RNase free. We eluted the RNA samples in 40-μL RNase-free H_2_O and then centrifuged at 11,000× *g* for 1 min. All RNA samples were stored at −80 °C.

### 4.12. Reverse Transcription-Polymerase Chain Reaction (RT-PCR)

Complementary DNA (cDNA) was synthesized using a High Capacity RNA-to-cDNA Kit (Thermo Fisher Scientific, Waltham, MA, USA). The reverse transcription (RT) reaction mixture contained: 2 × RT Buffer (10 μL), 20 × enzyme mix (1 μL), RNA sample (1 μg), and RNase-free H_2_O for a total volume 20 μL.

The reaction mixtures were mixed and spun down in the PCR tubes to collect all the samples in the bottom of the tubes. The reverse transcription in thermal cycling program in the PCR machine was set as: 37 °C for 60 min to start RT reaction, 95 °C for 5 min to inactive enzyme activity, then 4 °C for preservation. For long-term storage, we put the samples in a −20 °C freezer.

### 4.13. Quantitative PCR(Q-PCR)

After the RT was finished, we diluted the cDNA to 100 × with RNase-free water. For each sample, the following reaction mixture was added to one well of a 384-well Q-PCR plate: quantitative real-time PCR reaction mixture contains: cDNA (diluted with RNase-free water, 3.8 μL), 2.5 μM primer mix (forward and reverse, 1.2 μL), and 2 × SybrGreen Mix (5.0 μL), for a total volume of 10.0 μL.

The 2 × SybrGreen was added last because it is photosensitive. When the whole plate was ready, we covered it with an optical adhesive cover and smoothed out the bubbles with a sealing comb. The Q-PCR program was set as follows in an ABI HT-7900 (Thermo Fisher Scientific, Waltham, MA, USA) machine:Stage I: 50 °C—2 min; 95 °C—5 min; 4 °CStage II: 95 °C—10 minStage III (40 cycles): 95 °C—15 s; 60 °C—1 minStage IV: 95 °C—15 s; 60 °C—15 s; 95 °C—15 s

Dissociation Protocol: Start Temp: 60 °C

The resulting first-strand cDNA was used as a template for qualitative PCR performed in triplicate using the SYBR Green Q-PCR Master Mix Kit (Thermo Fisher Scientific, Waltham, MA, USA) using an ABI PRISM 7900 System (Thermo Fisher Scientific, Waltham, MA, USA). After normalization to internally controlled actin, the expression ratio between the experimental and control groups was calculated using the comparative Ct method. The relative expression ratio (fold change) was calculated based on △△Ct, △△Ct = (Ct _target_ – Ct _actin_)_treatment_ – (Ct _target_ – Ct _actin_)_control_, and fold change = 2^−△△Ct^. All experiments were performed in triplicate, and the mean values of three values are presented. At least three independent samples were used for Q-PCR, and the standard error was calculated and incorporated into the presented data as medians ± standard error. Differences among variables were assessed using a two-tailed Student’s *t*-test. A *p* < 0.05 was considered statistically significant and is shown as: *: 0.01 < *p* ≤ 0.05; **: 0.001 < *p* ≤ 0.01; and ***: *p* ≤ 0.001.

### 4.14. Tissue Preparation for Transplantation into Zebrafish Embryos

Human material from surgical resection specimens was obtained at the Wan-Fang Hospital (Taipei, Taiwan) according to the ethical guidelines reviewed by the Taipei Medical University Joint Institutional Review Board and after obtaining informed patient consent. Tissue samples from patients were cut into small pieces using scissors. Tissue pieces were then transferred to 15 mL containers with 3 mL isolation media (180 mL Dulbecco’s Modified Eagle Medium (DMEM) high glucose, 20 mL 100 mM HEPES (4-(2-hydroxyethyl)-1-piperazineethanesulfonic acid), 46 mL 5% BSA (bovine serum albumin) and collagenase (Thermo Fisher Scientific, Waltham, MA, USA) (50 μL of a 6 μg/μL stock solution (HBSS as solvent) for each 12 mL of isolation media), and then incubated the tissue in a water bath for 15 min at 37 °C. The supernatant was filtered through cell strainers (70 μm) and tissue pieces were cut further into smaller pieces using scissors. Tissue pieces were again incubated 15 min at 37 °C in 3 mL isolation media with collagenase. The cell suspension was centrifuged 5 min at 1500 rpm. The supernatant was discarded and cells re-suspended in isolation media. The described procedure was then repeated once. For injections, cells were stained with DiI or CellTrace™ CFSE/PBS (Thermo Fisher Scientific, Waltham, MA, USA) working solution (25 μM) for 15 min at 37 °C. The cells were centrifuged at 1500 rpm for 5 min, and washed with PBS. The supernatant was discarded, and the cells were resuspended in PBS and diluted at 4.35 × 10^4^ cells/μL.

### 4.15. Xenotransplantation and Imaging for Monitoring Drug Response

The fertilized eggs of *Tg(fli1:EGFP)* zebrafish were incubated at 28 °C in E3/PTU solution and raised under standard zebrafish laboratory conditions. Zebrafish embryos two days post-fertilization were dechorionated and anesthetized with 0.016% tricaine methanesulfonate (MS-222) (MilliporeSigma, St. Louis, MO, USA) before microinjection. The 293T/EDN1 cells collected in PBS were labeled in vitro with CM-Dil (red fluorescence) (Vybrant; Invitrogen, Carlsbad, CA, USA). Each injection volume of 4.6 nL contained about 200 cells and implanted into each yolk of 2 dpf zebrafish embryo via a glass capillary using a Nanoject II^TM^ nanoliter injector (Drummond Scientific, Broomall, PA, USA). After injection, zebrafish embryos were washed once with E3/PTU solution, and incubated for 1 h at 28 °C and checked for the presence of fluorescent cells at 2 h post-transplantation. After 24 h post-transplantation, the zebrafish were treated with BPR1J419S1 and BPR1J420S1 separately, and we observed the cell proliferation and migration abilities in the following 3–5 days using a LEICA DM IRB fluorescent microscope (Leica Microsystems Inc., Buffalo Grove, IL, USA)

Xenotransplantation was performed in 2-dpf embryos. For microinjection of human tumor cells to the fish embryos, we followed these protocols: fish embryos were treated with mild protease to remove the protective chorions at 1 dpf. Approximately 200 cells per 4.6 nL were injected into 2 dpf embryos. The injection site was the yolk. The Nanoject II system (Drummond Scientific Company, Broomall, PA, USA) was used with capillaries of 3.5′ (Drummond #3-000-203-G/X; Drummond Scientific Company, Broomall, PA, USA) pulled by a flaming/brown micropipette puller model P-87 (Sutter Instrument Co., Novato, CA, USA) in a fixed parameter (heat: 607, pull: 180, velocity: 150, delay time: 100 and pressure: 500). The end of the pulled needle was then removed using watchmaker forceps to produce an opening, and ground using a EG-400 Microgrinder (NARISHIGE Tokyo, Japan) to create a 40-degree grinding plane and a 125-μm diameter section opening. Injections were performed under an XL-720 microscope (SAGE VISION, Bala Cynwyd, PA, USA). After injection, the embryos were cultured in E3/PTU buffer (E3: 5 mM NaCl, 0.17 mM KCl, 0.33 mM CaCl_2_, and 0.33 mM MgSO_4_) in petri dishes in temperature-controlled incubators (28 °C). The embryos with leaking injected cells were removed at 2 h post-injection (hpi) and placed in the temperature gradient incubator that automatically increased 1 °C every six hours, and reached 37 °C in two days.

#### 4.15.1. Manual Imaging

We took zebrafish with 1 × tricaine (0.016 mg/mL) and images were captured using an OLYMPUS SZX10 fluorescence microscope coupled with a DP71 and U-RFL-T camera (OLYMPUS, Tokyo, Japan) at 1 dpi. The embryos were then placed in a 96-well plate, with 280 μL per well containing one zebrafish embryo. The drugs were dissolved in E3/PTU buffer and 1% DMSO as control. At 2 dpi, we changed the media with fresh solution and control. Two days after drug treatment (3 dpi), the images of the same embryo were captured again. The tumor cells in embryos were observed, and the migration and proliferation ability of the injected cells were recorded and analyzed using ImageJ software (National Institutes of Health (NIH), Bethesda, MD, USA).

#### 4.15.2. Automatic Imaging

The 1-dpi embryos were placed in a 96-well half-area plate with 150 μL (0.016 mg/mL) tricaine per well to take images using auto-machine (ImageXpress^R^ Micro, Molecular Devices, Sunnyvale, CA, USA). Then, 180 μL per well containing one zebrafish embryo was selected for drug treatment. The drugs were dissolved in E3/PTU buffer and 1% DMSO as control. At 2 dpi, we changed the media with fresh solution and control. We captured images at 3 dpi with Automated High Throughput Screening System (HTS) (BMG LABTECH, Allmendgrün 8, Ortenberg, Germany). The images were analyzed using MetaXpress MDCStore 2.3 software (Molecular Devices, Sunnyvale, CA, USA).

### 4.16. Embryonic Toxicity Test

Zebrafish embryos were harvested at 4 hpf and incubated at 28 °C for the duration of the experiments. Forty embryos were placed into each well of 6-well polystyrene tissue culture plates. We added 5 mL of a different concentrations of compounds to each well. The buffer was renewed every day throughout the experiment. The morphology and survival rate were observed at different time points: 12, 24, 36, 48, 72, 96, and 120 h. The images were captured using an Olympus SZX10 stereo fluorescence microscope coupled with a DP71 digital Charge Coupled Device (CCD) camera (OLYMPUS, Tokyo, Japan).

### 4.17. Hepatotoxicity Test

*EGFP-mCherry* embryos collection and incubation conditions were previously described in Section 4.4. At about 3 dpf, 50 embryos were distributed into 10 mL chemical solution/well (chemical/E3 solution) in 6-well plates until 5 dpf and the chemical solution was replaced every day. At 5 dpf, embryos were anesthetized with tricaine (0.016%) and images of 8 to 10 randomly chosen embryos per well were taken with a ZEISS AxioCam MRc (ZEISS, Oberkochen, Germany). We captured three different images per embryo: one with automatic exposure time for the clearest view, one with fixed exposure time to capture RFP (red fluorescent protein) intensity below saturation for intensity measurement and comparison, and the one with sufficient exposure time to show the whole liver region for size measurement. ImageJ software was then used to quantify intensity of RFP and liver size. Average RFP intensity in the liver was calculated and compared within the same group of lateral view fry under the same magnification and fixed exposure time.

### 4.18. Survival Test

*Tg(fli1:EGFP)* embryos were used in the survival assay. At 3 dpf, 20 embryos were placed into one well of the 6-well plates with 2 mL E3 medium supplement with drugs. The DMSO control and three different drugs, 419S1, 420S1, and Sorafenib, were serially diluted to determine the survival rate. Two days after exposure, the embryos were counted and the survival curves were measured.

### 4.19. Statistical Analysis

The statistical analysis of the results was performed using Prism 8 (GraphPad Software, San Diego, CA, USA) and two-tailed Student’s *t*-test was applied. In all statistical analyses, a *p*-value < 0.05 was considered to be statistically significant and is shown as: *: 0.01 < *p* ≤ 0.05; **: 0.001 < *p* ≤ 0.01; and ***: *p* ≤ 0.001.

## 5. Conclusions

Zebrafish are an excellent model for drug screening; many drug screening platforms using zebrafish embryos have been established. Our laboratory established the zebrafish HCC models, the Institute of Biotechnology and Pharmaceutical Research (IBPR) has small molecular libraries, and Wen-Fang Hospital provided the patients’ specimens. To establish a high-throughput platform for drug screening, we started with anti-angiogenesis using zebrafish larva. We integrated the strengths of our transgenic zebrafish and small molecular libraries with the zebrafish drug screening platform. With the integrated efforts and advantages of the zebrafish platform, we found more effective and safer drugs for anti-HCC treatment. We developed a high-throughput drug screening platform to identify novel and safe anti-HCC therapeutic drugs using zebrafish.

## Figures and Tables

**Figure 1 cancers-11-00739-f001:**
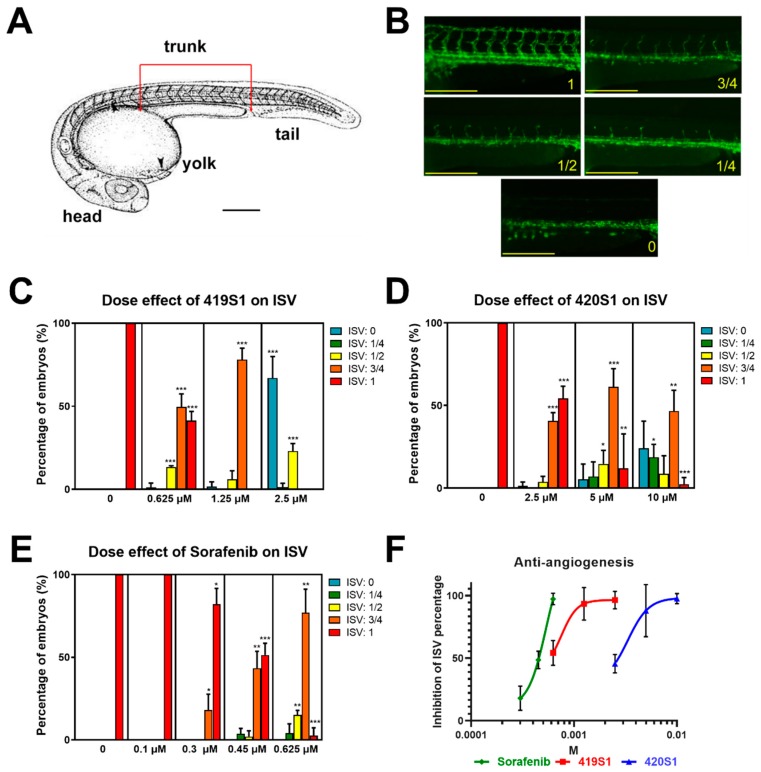
Titrations of 419S1, 420S1, and Sorafenib, and determination of the half maximal inhibitory concentration (IC_50_) for anti-angiogenesis. (**A**) Schematic illustration of the zebrafish embryo at two days post fertilization (dpf), indicating the trunk region for measuring the length of intersegmental vessels (ISVs). (**B**) Representative images of the 2 dpf embryos exposed to compounds for one day with the lengths of ISVs reaching the dorsal longitudinal anastomotic vessel (DLAV) fully (1), three quarters (3/4), halfway (1/2), one-quarter (1/4), or none (0). Scale bar of **A** and **B**: 0.2 mm. (**C**) Bar chart (mean and S.E.M.) showing a significant reduction in the length of ISVs after one day immersion with 419S1 at three concentrations (0.625, 1.25, and 2.5 μM). (**D**) Quantification of ISVs showed a significant inhibition of angiogenesis after 420S1 treatment at three concentrations (2.5, 5, and 10 μM). (**E**) Exposure to Sorafenib lead to a significant reduction in the length of ISVs at four concentrations (0.1, 0.3, 0.45, and 0.625 μM). (**F**) The dose-response fitting curve of 419S1, 420S1, and Sorafenib was generated by Prism8 (GraphPad Software, San Diego, CA, USA). Inhibition of ISV percentage is the percentage of embryos combined with all form of shortened ISVs. *: 0.01 < *p* ≤ 0.05; **: 0.001 < *p* ≤ 0.01; ***: *p* ≤ 0.001.

**Figure 2 cancers-11-00739-f002:**
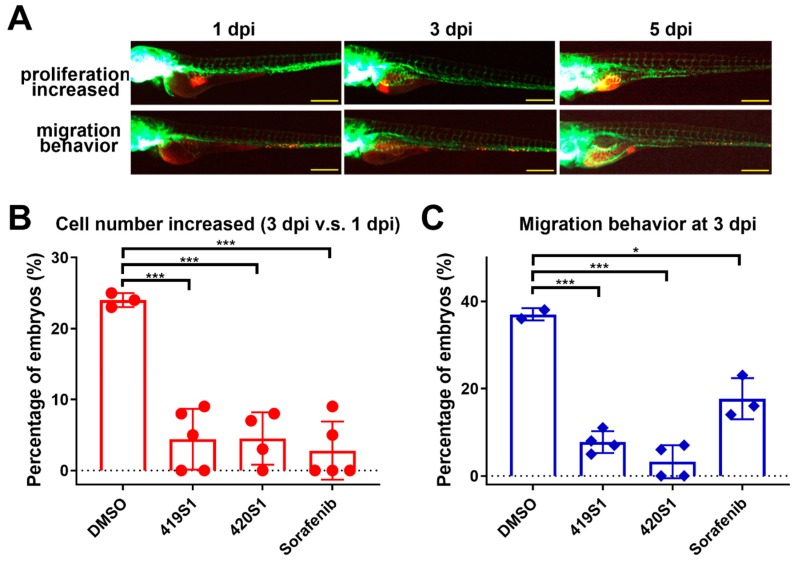
419S1, 420S1, and Sorafenib exhibit anti-proliferation and anti-migration abilities in a xenotransplantation assay. (**A**) Schematic diagrams of proliferation and migration changes at one, three, and five days post-injection (dpi). The red fluorescence was the CM-DiI-labeled 293T cells overexpressing EDN1 oncogene microinjected into 2 dpf *Tg(fli:EGFP)* zebrafish embryos where blood vessels express green fluorescence. One dpi images of embryos carrying 293T/EDN1 were captured, compounds were added to the solution, and images of embryos were captured two days after drug treatment at three dpi, or four days of drug treatment at five dpi. Scale bar: 0.2 mm. (**B**) Dot-plot (mean and S.E.M.) showing 419S1, 420S1, and Sorafenib treatment significantly decreased the percentage of embryos with tumor cell proliferation. (**C**) Dot-plot (mean and S.E.M.) showing 419S1, 420S1, and Sorafenib treatment significantly decreased the percentage of embryos with tumor cell migration behavior compared to the dimethyl sulfoxide (DMSO) control. *: 0.01 < *p* ≤ 0.05; **: 0.001 < *p* ≤ 0.01; ***: *p* ≤ 0.001.

**Figure 3 cancers-11-00739-f003:**
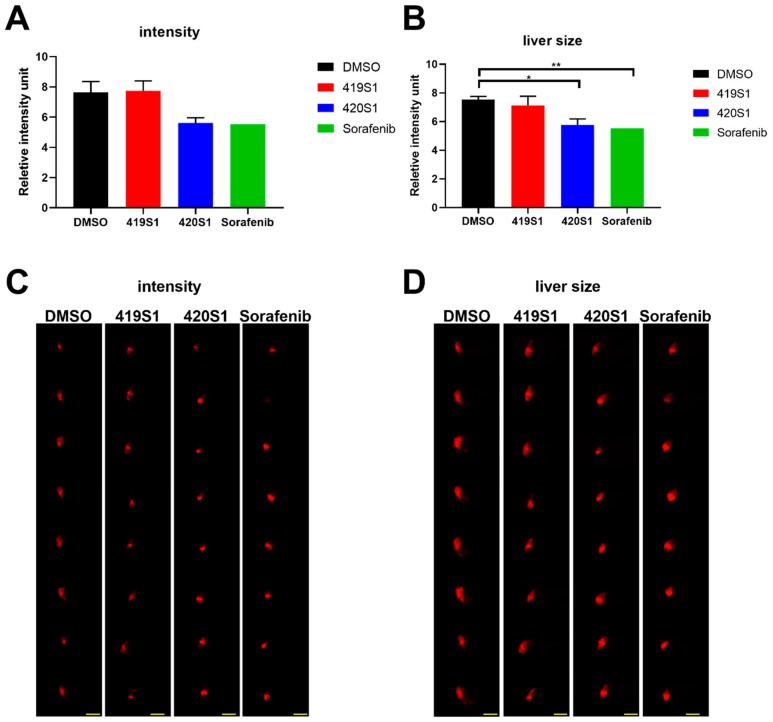
Liver red fluorescence and liver size *Tg(fabp10a:EGFP-mCherry)* embryos treated with 419S1, 420S1, and Sorafenib for two days starting at three dpf. (**A**) *Tg(fabp10a:EGFP-mCherry)* embryos treated with 420S1 and Sorafenib showed significant decreases in liver fluorescence intensity compared to DMSO control; 419S1 does not reduce the fluorescence intensity. (**B**) *Tg(fabp10a:EGFP-mCherry)* embryos treated with 420S1 and Sorafenib had significant decreases in liver size compared to the DMSO control; 419S1 did not reduce liver size. (**C**) Representative images with fixed exposure time to capture the liver fluorescence intensity of embryos treated with DMSO, 419S1, 420S1, and Sorafenib. (**D**) Representative images with sufficient exposure time to show the whole liver size of embryos treated with DMSO, 419S1, 420S1, and Sorafenib. Scale bar of **C** and **D**: 0.25 mm. *: 0.01 < *p* ≤ 0.05; **: 0.001 < *p* ≤ 0.01; ***: *p* ≤ 0.001.

**Figure 4 cancers-11-00739-f004:**
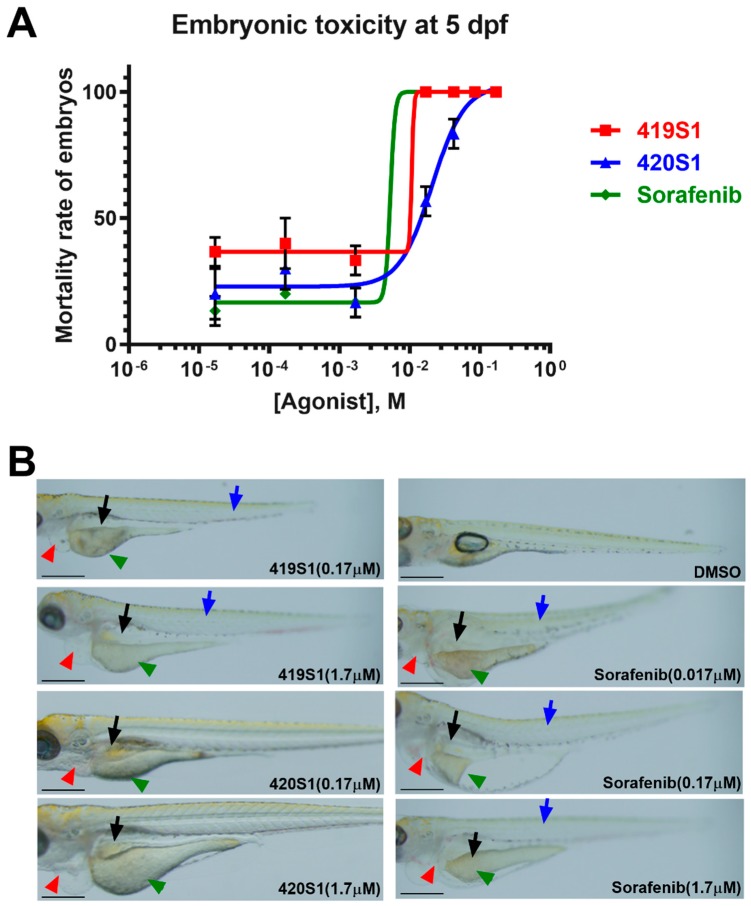
The dose–response fitting mortality curve of wide-type embryos treated with 419S1, 420S1, and Sorafenib for five days, starting six hours post-fertilization. (**A**) Embryos treated with Sorafenib had a significantly higher mortality rate compared to 419S1 and 420S1. (**B**) Representative images of wild-type embryos treated with 419S1, 420S1, and Sorafenib at specified concentrations for five days. Abnormal embryonic phenotypes are shown with arrows. Red arrow: pericardial edema; green arrow: retarded yolk sac reabsorption; black arrow: retarded swim bladder inflation; blue arrow: spinal curvature and/or short body length. Scale bar: 0.2mm.

**Figure 5 cancers-11-00739-f005:**
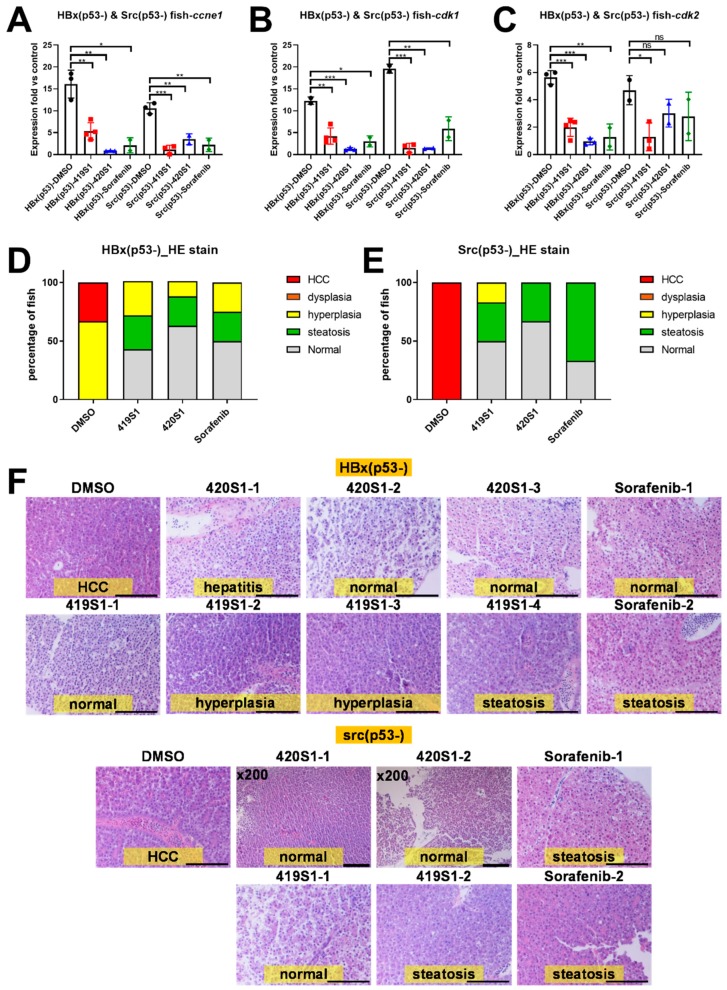
Treatment with 419S1 and 420S1 blocked hepatocellular carcinoma (HCC) formation in *Tg(fabp10a:HBx, p53^−^)* and *Tg(fabp10a:src, p53^−^)* transgenic zebrafish. (**A**–**C**) Dot-plot of the relative expression fold of cell-cycle-related genes *ccne1*, *cdk1*, and *cdk2* after drug treatment compared to control fish. DMSO-treated *Tg(fabp10a:HBx, p53^−^)* and *Tg(fabp10a:src, p53^−^)* transgenic fish exhibited high levels of *ccne1*, *cdk1,* and *cdk2* expression compared to control fish. DMSO- (black circle), 419S1- (red square), 420S1- (blue triangle), and Sorafenib (green diamond)-treated *Tg(fabp10a:HBx, p53^−^)* and *Tg(fabp10a:src, p53^−^)* transgenic fish demonstrated significantly decreased *ccne1* expression compared to DMSO. (**D**,**E**) Percentage of fish with various histopathological features revealed by hematoxylin and eosin (H&E) staining after different drug treatments for one month. Red indicates HCC, orange denotes dysplasia, yellow represents hyperplasia, green denotes steatosis, and grey is normal hepatocyte. (**F**) Representative images of H&E stain (400 ×) after one month of 419S1, 420S1, or Sorafenib treatment. Top panels are *Tg(fabp10a:HBx, p53^−^)* and bottom panels are *Tg(fabp10a:src, p53^−^)* transgenic fish. Scare bar: 100 μm. *: 0.01 < *p* ≤ 0.05; **: 0.001 < *p* ≤ 0.01; ***: *p* ≤ 0.001.

**Figure 6 cancers-11-00739-f006:**
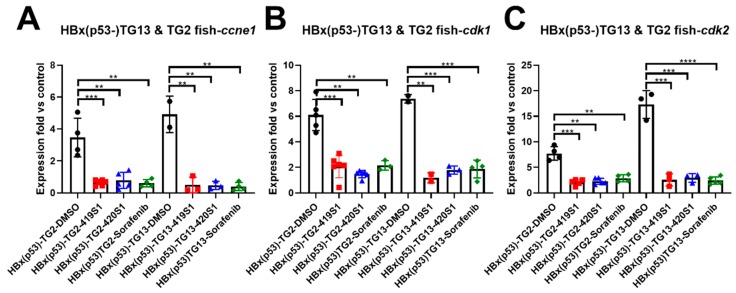
The treatment of 419S1 and 420S1 blocked HCC formation in *Tg(fabp10a:HBx,p53^−^)*-TG13 and TG2 transgenic zebrafish revealed by Q-PCR analysis or proliferation markers. (**A**) Dot-plot of relative expression fold of cell-cycle-related gene *ccne1* after drug treatment compared to control fish. DMSO-treated *Tg(fabp10a:HBx,p53^−^)* fish exhibited high levels of *ccne1* expression compared to control fish. DMSO- (black circle), 419S1- (red square), 420S1- (blue triangle), and Sorafenib (green diamond)-treated *Tg(fabp10a:HBx,p53^−^)* fish demonstrated significantly decreased *ccne1* expression compared to DMSO. (**B**) Expression level of cell-cycle-related genes *cdk1* for 419S1 and 420S1 in TG13 and TG2 of *Tg(fabp10a:HBx,p53^−^)*. (**C**) Expression level of cell-cycle-related genes *cdk2* for 419S1 and 420S1 in TG13 and TG2 of *Tg(fabp10a:HBx,p53^−^)*. **: 0.001 < *p* ≤ 0.01; ***: *p* ≤ 0.001.

**Figure 7 cancers-11-00739-f007:**
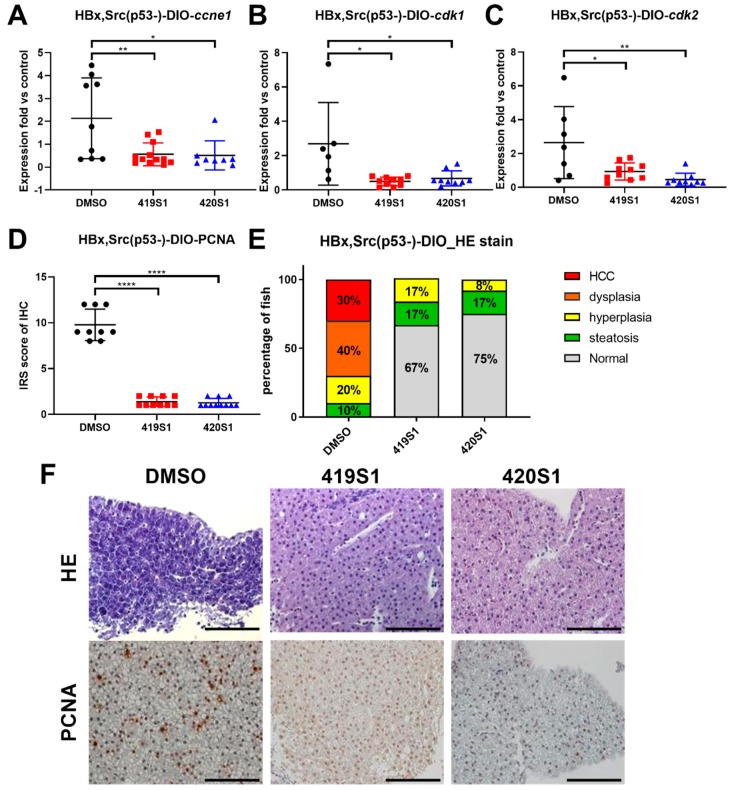
The treatment of 419S1 and 420S1 blocked HCC formation in *Tg(fabp10a:HBx,src, p53^−^)* transgenic zebrafish. (**A**–**C**) Dot-plot of relative expression fold of cell-cycle-related genes *ccne1*, *cdk1*, and *cdk2* after 419S1 and 420S1 oral gavage for one month in *Tg(fabp10a:HBx,src, p53^−^)* compared to control fish. DMSO-treated *Tg(fabp10a:HBx,src, p53^−^)* fish exhibited high levels of *ccne1*, *cdk1*, and *cdk2* expression compared to control fish. DMSO- (black circle), 419S1- (red square), and 420S1 (blue triangle)-treated *Tg(fabp10a:HBx,src, p53^−^)* fish showed significantly decreased *ccne1*, *cdk1*, and *cdk2* expression compared to DMSO. (**D**) Dot-plot of relative expression fold of immunoreactive score (IRS), which was obtained by multiplying the intensity grade by the percentage rating from PCNA IHC stain. (**E**) Percentage of fish with various histopathological features revealed by H&E stain after different drug treatments for one month. Red indicates HCC, orange denotes dysplasia, yellow represents hyperplasia, green denotes steatosis, and grey is normal hepatocyte. (**F**) Representative images of H&E stain (400 ×) and PCNA IHC staining after 419S1 and 420S1 oral gavage for one month in *Tg(fabp10a:HBx,src, p53^−^)* compared to DMSO control. Scare bar: 100 μm. *: 0.01 < *p* ≤ 0.05; **: 0.001 < *p* ≤ 0.01; ****: *p* ≤ 0.0001.

**Figure 8 cancers-11-00739-f008:**
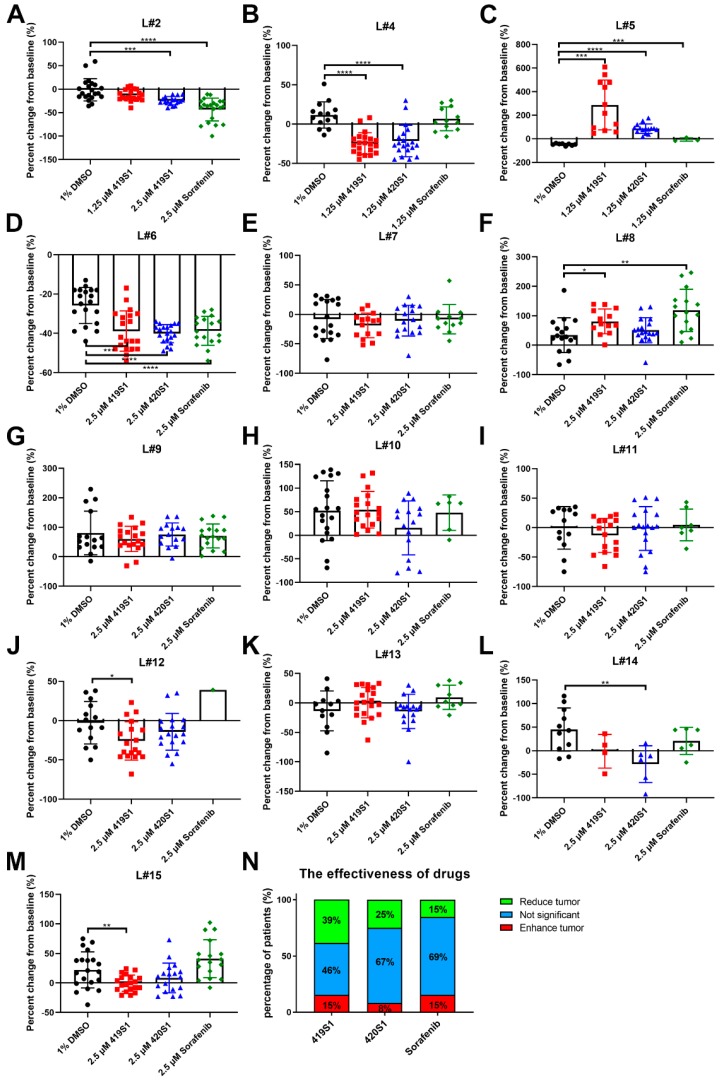
419S1, 420S1 and Sorafenib exhibit anti-proliferation and anti-migration abilities in a liver cancer patient-derived-xenograft (PDX). (**A**–**M**) Embryos carried tumor cells from different patients (labeled on top) were labeled with fluorescence dye and injected into 2 dpf zebrafish embryos, and treated with DMSO (black circle), 419S1 (red square), 420S1 (blue triangle), and Sorafenib (green diamond) for two days starting one day post-injection of the tumor cells. (**N**) Different patients’ tumors responded differently to the drugs. The effectiveness of the drugs was summarized and showed 419S1 as the most effective. Red indicates proliferation increased, blue denotes proliferation did not change after drug treatment, and green denotes proliferation decreased after drug treatment. *: 0.01 < *p* ≤ 0.05; **: 0.001 < *p* ≤ 0.01; ***: 0.0001 < *p* ≤ 0.001; ****: *p* ≤ 0.0001.

**Table 1 cancers-11-00739-t001:** The primer sequence of Q-PCR for cell- cycle-related genes: *ccne1*, *cdk1*, and *cdk2*.

Gene Name	Primer Name	Start	Sequence (5′ to 3′)	Accession Number	Size (bp)
*ccne1*	Q-*ccne1*-F	371	TCCCGACACAGGTTACACAA	NM_130995.1	201
	Q-*ccne1*-R	571	TTGTCTTTTCCGAGCAGGTT		
*cdk1*	Q-*cdk1*-F	779	CTCTGGGGACCCCTAACAAT	NM_212564.2	200
	Q-*cdk1*-R	978	CGGATGTGTCATTGCTTGTC		
*cdk2*	Q-*cdk2*-F	794	CAGCTCTTCCGGATATTTCG	NM_213406.1	199
	Q-*cdk2*-R	992	CCGAGATCCTCTTGTTTGGA		

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
