# Peer review of "Identification of Novel Anti-Liver Cancer Small Molecules with Better Therapeutic Index Than Sorafenib via Zebrafish Drug Screening Platform"

_cancers, 2019, doi:10.3390/cancers11060739_

Reviewer 1 Report

The manuscript is of considerable scientific interest and worthy of publication. I report only an error in the caption of figure 1 letter E.

Author Response

Reviewer #1: English language and style are fine/minor spell check required.

Comments: The manuscript is of considerable scientific interest and worthy of publication. I report only an error in the caption of figure 1 letter E.

Thanks to Reviewer #1, we have corrected the figure 1E caption, and also have English Editing Company edited the entire manuscript.

Reviewer 2 Report

In this study, authors attempt to investigate the zebrafish drug screening platform as well as compare the therapeutic results of two multiple tyrosine kinase inhibitors, 419S1 and 420S1 with Sorafenib. It was observed that all three compounds showed anti-angiogenesis ability by immersed the fli1: EGFP transgenic embryos, the IC50 (half inhibition concentration) was obtained. The 419S1 was noted to have lower hepatoxicity and embryonic toxicity than 420S1 and Sorafenib.  Authors found the therapeutic index (LC50/IC50) for 419S1 to be higher than the ones for Sorafenib and 420S1. Additionally, when the compounds were injected to adult transgenic zebrafish with HCC, pathological features were cured, while the expression levels of cell-cycle-related genes and protein levels of proliferation marker were reversed. Authors also found that in the patient-derived-xenograft assay, the effectiveness of Sorafenib to be better than of 419S1 and 420S1 to prevent liver cancer. Authors concluded that using zebrafish platform, more efficacious and safer drugs for anti-HCC may be developed.

Comments:

Overall it is a good, well-written study. All experiments are well perfumed, well reported. However, I have two suggestions;

1-      In the introduction section, Line 43, authors write, ‘’ Risk factors for HCC including 43 hepatitis B and C virus infection, alcohol and non-alcoholic liver diseases’’ In my opinion, there are some other risk factors such as iron storage diseases, aflatoxin are also considered risk factor and may be included.

2-      Authors mention ‘personalized medicine’ in abstract and also in other sections, but no mention of it in the introduction, which in my opinion is important.

Author Response

Reviewer #2: Moderate English changes required.

Comments: Overall it is a good, well-written study. All experiments are well perfumed, well reported. However, I have two suggestions;

1- In the introduction section, Line 43, authors write, ‘’ Risk factors for HCC including 43 hepatitis B and C virus infection, alcohol and non-alcoholic liver diseases’’ In my opinion, there are some other risk factors such as iron storage diseases, aflatoxin are also considered risk factor and may be included.

2- Authors mention ‘personalized medicine’ in abstract and also in other sections, but no mention of it in the introduction, which in my opinion is important.

Thanks to the advice from Reviewer #2, we have English Editing Company edited the entire manuscript. We also edited the Introduction section adding other risk factors for HCC, and also added a paragraph for personalized medicine in the introduction.

Reviewer 3 Report

In this study, the authors established a zebrafish drug screening platform and compared the therapeutic effect of two multiple tyrosine kinase inhibitors, 419S1 and 420S1 with Sorafenib. The authors found that all compounds exhibited anti-angiogenesis ability, however, the 419S1 exhibited lower hepatoxicity and embryonic toxicity than 420S1 and Sorafenib. Moreover, the patient-derived-xenograft zebrafish model showed that the effectiveness of 419S1 and 420S1 to prevent liver cancer proliferation is better than Sorafenib, showing these compounds might be useful in clinical trial for future development.

This study is well-organized, and done with good quality. The finding of this study is interesting and may contribute to HCC treatment. In general, the interpretation of the data is reasonable; however, the following issues need to be addressed.

Comments:

1. There are still some differences between zebrafish and human physiology. It is recommended that the authors further validate the findings in the mouse xenograft model to obtain more powerful evidence.

2. The legends of all the figures are too brief. The details about procedures for each experiment should be added.

3.  Scale bar in all microscopy should be added.

4. The manuscript contains many grammatical and syntax errors and must be carefully edited.

Author Response

Reviewer #3: English language and style are fine/minor spell check required.

Comments:

1. There are still some differences between zebrafish and human physiology. It is recommended that the authors further validate the findings in the mouse xenograft model to obtain more powerful evidence.

2. The legends of all the figures are too brief. The details about procedures for each experiment should be added.

3. Scale bar in all microscopy should be added.

4. The manuscript contains many grammatical and syntax errors and must be carefully edited.

Thanks to the suggestions from Reviewer #3, we have English Editing Company edited the entire manuscript. We have re-written the legends for all figures with detail explanation about the procedure for each experiment. We also added the scale bar for all the images of microscopy, and the grammatical and syntax errors have been carefully edited by MDPI English Editing Service.

Although there are some differences between zebrafish and human physiology, the physiology of mouse also different from human. Moreover, immune deficient mouse will be used for xenograft, which is very different from human. In this study, we injected the small molecules into blood, or oral-fed the small molecules to three different adult transgenic fish HCC models, we proved small molecules 419S1 and 420S1 exhibited the therapeutic effects, we believe those are powerful evidences compared to mouse xenograft. We also used patient-derived-xenograft in zebrafish embryos and provided evidences for the anti-tumor proliferation effects. Mouse xenograft models had been tested in our lab, the process is much time-consuming and more expensive than fish.

Round  2

Reviewer 3 Report

The authors have addressed all questions raised during the first reviewing procedure. I have found the manuscript to be much improved and recommended acceptance.